# CCN2 Binds to Tubular Epithelial Cells in the Kidney

**DOI:** 10.3390/biom12020252

**Published:** 2022-02-03

**Authors:** Sandra Rayego-Mateos, José Luis Morgado-Pascual, Carolina Lavoz, Raúl R. Rodrigues-Díez, Laura Márquez-Expósito, Antonio Tejera-Muñoz, Lucía Tejedor-Santamaría, Irene Rubio-Soto, Vanessa Marchant, Marta Ruiz-Ortega

**Affiliations:** 1Molecular and Cellular Biology in Renal and Vascular Pathology, IIS-Fundación Jiménez Díaz, Universidad Autónoma Madrid, Av Reyes Católicos 2, 28040 Madrid, Spain; srayego@fjd.es (S.R.-M.); laura.marqueze@quironsalud.es (L.M.-E.); antonio.tejera@quironsalud.es (A.T.-M.); lucia.tejedor@quironsalud.es (L.T.-S.); irene.rubios@quironsalud.es (I.R.-S.); vmarchant.hernandez@gmail.com (V.M.); 2Red de Investigación Renal (REDinREN), Av. de Monforte de Lemos, 5, 28029 Madrid, Spain; rrodriguez@fjd.es; 3Maimonides Biomedical Research Institute of Cordoba (IMIBIC), Hospital Universitario Reina Sofía, 14004 Cordoba, Spain; jose.morgado@uco.es; 4Department of Cell Biology, Physiology and Immunology, University of Cordoba, 14071 Cordoba, Spain; 5Division of Nephrology, School of Medicine, Universidad Austral Chile, Valdivia 5090000, Chile; carolavoz@gmail.com; 6Translational Immunology Laboratory, Health Research Institute of Asturias (ISPA), 33011 Oviedo, Spain

**Keywords:** CCN2, CTGF, EGFR, kidney damage

## Abstract

Cellular communication network-2 (CCN2), also called connective tissue growth factor (CTGF), is considered a fibrotic biomarker and has been suggested as a potential therapeutic target for kidney pathologies. CCN2 is a matricellular protein with four distinct structural modules that can exert a dual function as a matricellular protein and as a growth factor. Previous experiments using surface plasmon resonance and cultured renal cells have demonstrated that the C-terminal module of CCN2 (CCN2(IV)) interacts with the epidermal growth factor receptor (EGFR). Moreover, CCN2(IV) activates proinflammatory and profibrotic responses in the mouse kidney. The aim of this paper was to locate the in vivo cellular CCN2/EGFR binding sites in the kidney. To this aim, the C-terminal module CCN2(IV) was labeled with a fluorophore (Cy5), and two different administration routes were employed. Both intraperitoneal and direct intra-renal injection of Cy5-CCN2(IV) in mice demonstrated that CCN2(IV) preferentially binds to the tubular epithelial cells, while no signal was detected in glomeruli. Moreover, co-localization of Cy5-CCN2(IV) binding and activated EGFR was found in tubules. In cultured tubular epithelial cells, live-cell confocal microscopy experiments showed that EGFR gene silencing blocked Cy5-CCN2(IV) binding to tubuloepithelial cells. These data clearly show the existence of CCN2/EGFR binding sites in the kidney, mainly in tubular epithelial cells. In conclusion, these studies show that circulating CCN2(IV) can directly bind and activate tubular cells, supporting the role of CCN2 as a growth factor involved in kidney damage progression.

## 1. Introduction

Cellular communication network-2 (CCN2), also called connective tissue growth factor (CTGF), is a member of the CCN family and shares many properties with other CCN proteins [1,2]. Intensive research in the last years has unraveled the complexity of the CCN family. All CCN family members are matricellular proteins, key components of the extracellular matrix (ECM), that contain multiple ligand and receptor binding sites and are actively involved in cellular communication and signaling. However, key differences between them have been described, with CCN2 being especially relevant in renal diseases [3,4].

CCN2 is a complex matricellular protein that exerts multiple cellular actions based on its special structure and cellular localization [3,4,5,6]. CCN2 has four functional modules that can be cleaved by proteases, releasing active fragments. Among them, the carboxyl-terminal cysteine knot module 4 (here named CCN2(IV)) shares many responses with the full-length CCN2 [3,4]. Numerous preclinical studies using the recombinant CCN2 complete protein, or some of its fragments, have described a wide diversity of cellular responses and activation of molecular pathways [7]. One of these cellular responses is cell growth, in which CCN2 participates in the regulation of proliferation, apoptosis, and migration [8]. In addition, CCN2 can also promote phenotype changes, as described in T lymphocytes differentiation towards Th17 subtype [9], in the induction of cellular senescence [10,11], and in the processes of epithelial/endothelial to mesenchymal transitions (EMT/EndMT, respectively) [12,13]. CCN2 is also involved in angiogenesis, cancer, inflammation, and fibrogenesis [14,15,16,17,18]. In fact, CCN2 is considered a key fibrotic marker, commonly used in preclinical models [18,19]. Thus, based on these investigations, functions of CCN2 have been widely studied, finally according to it as a growth factor and a proinflammatory cytokine. However, more recent studies using mainly conditional CCN2 knockout mice have remarked the importance of CCN2 as a matricellular protein implicated in cell-to-cell communication and response to micro-environmental signaling [3,17]. These studies have shown that CCN2 exerts key cellular responses in physiological and pathological conditions. Thus, CCN2 acts as an ECM component, involved in the regulation of other ECM components and therefore contributing to matrix remodeling [20], tissue stiffness [1,17], and pathological ECM accumulation in fibrotic disorders, including kidney fibrosis [21], and more recently, participating in vascular wall homeostasis or adaptive remodeling for prevention of structural damage, especially in the context of hypertension and inflammation [22]. 

In human chronic kidney disease (CKD) of diverse etiology, kidney CCN2 overexpression has been found correlated with cellular proliferation and ECM accumulation, both in the glomerular and interstitial areas [23,24,25,26]. Initial studies done in cultured renal cells demonstrated that CCN2 expression is upregulated by many factors involved in renal damage, including transforming growth factor-β (TGF-β), angiotensin II, elevated glucose concentrations, and cellular stress [18,24,25,27]. Interestingly, in several independent studies, circulating or urinary levels of CCN2 have been proposed as a risk biomarker of human diabetic nephropathy and other forms of CKD [19,28,29]. Additionally, CCN2 has also been proposed as a potential therapeutic target for renal diseases. Preclinical studies have shown that inhibition of endogenous CCN2 by antisense oligonucleotides or gene silencing slows disease progression in experimental diabetic nephropathy, unilateral ureteral obstruction, and nephrectomized transgenic mice overexpressing TGF-β1 [19,30,31,32]. More recently, studies in CCN2 conditional deficient mice have demonstrated the role of CCN2 in renal fibrosis, cell growth arrest, oxidative stress, and senescence [11,21].

The CCN2 signaling receptor is still controversial. Different studies have shown that integrins are involved in CCN2 downstream signaling responses [7], which turns out to be relevant in some disorders such as cancer. In this regard, CCN2 can also bind to non-integrin receptors [16,33], such as low-density lipoprotein receptor-like protein 1 (LRP-1), receptor activator of NF-κB (RANK), osteoprotegerin, fibroblast growth factor receptor, tyrosine kinase receptor of nerve growth factor (TrkA), cation-independent mannose-6-phosphate (M6P) receptor, and epidermal growth factor receptor (EGFR) [1,33,34], all of them multi-ligand receptors. In addition, CCN2 can bind to several growth factors, being able to modify their responses [7,27]. These multiple receptors and ligand binding sites are a characteristic shared with matricellular proteins and other CCN proteins. However, the downstream functional consequences depend on the environment and physiopathological situation. In the kidney context, two potential CCN2 receptors have been described in in vitro studies. TrkA mediates CCN2 responses in cultured mesangial [35] and tubular epithelial [34] cells, whereas EGFR is involved in tubular epithelial cells responses, including partial EMT phenotype changes [21]. 

The EGFR pathway activation regulates several cellular functions, including development, angiogenesis, migration, and ECM production [7,36]. EGFR participates in cancer development and progression in many human malignancies, where it is overexpressed, dysregulated, or mutated [36,37,38]. In the kidney, EGFR signaling participates in embryonic development and controls renal electrolyte homeostasis [39]. Experimental studies have shown that genetic or pharmacological EGFR inhibition ameliorates renal damage progression [40,41,42], whereas in acute kidney injury, EGFR activation can accelerate renal recovery [43]. EGFR binds to multiple ligands, being relevant in kidney diseases, mainly EGF, transforming growth factor (TGF)-α, heparin-binding EGF-like growth factor (HB-EGF) [7,40,42], amphiregulin [44], and, as we have previously described, CCN2 [34]. Surface plasmon resonance analysis demonstrated that EGFR binding was located at the C-terminal module of CCN2 [34]. In previous studies, using an EGFR kinase inhibitor, we have described that CCN2-induced profibrotic responses in the kidney [21], endothelial dysfunction, and vascular inflammation [45] were mediated by EGFR signaling pathway activation. Therefore, considering all previous information provided, the aim of the present study was to localize CCN2 binding in vivo in the kidney, using the C-terminal module CCN2(IV) labeled with a fluorophore (Cy5) and confocal microscopy analysis, evaluating whether EGFR is a functional receptor for CCN2.

## 2. Materials and Methods

### 2.1. Experimental Model of CCN2-Induced Renal Damage

Animal procedures were carried out in 3-month-old male C57BL/6 mice according to the guidelines for animal research in the European Community, with prior approval by the Ethics Committee of the Health Research Institute of the IIS-Fundación Jiménez Díaz.

To localize CCN2 binding sites, Cy5-labeled CCN2(IV) was used, and two different mice models were done: intra-renal injection or intraperitoneal injection (i.p.) of CCN2(IV) Cy5-labeled. Choice of CCN2(IV) dose was based on previous studies done with systemic CCN2(IV) administration by i.p. in mice [34]. Different administration routes were used to evaluate the effect of local vs. systemic administration of recombinant protein as well as binding stability of the fluorescent compound. The purity of CCN2(IV) (endotoxin levels < 0.01) was also evaluated by matrix-assisted laser desorption/ionization-time of flight (MALDI-TOF) mass spectrometry (data not shown).

*Intra-renal injection model:* This model of renal parenchymal administration of compounds was performed under isoflurane anesthesia, as described elsewhere [46,47]. Cy5-CCN2(IV) (2.5 ng/g of body weight) was injected into the right kidney of mice; using the other (contralateral) kidney as a control, the analysis was performed at different times (10, 15, and 30 min). As additional control, a sham group consisting of mice injected with saline in the right kidney was established; in this group, the contralateral kidney was left untreated.

*Intraperitoneal injection model:* Mice received a single intraperitoneal injection of Cy5-labeled CCN2(IV) dissolved in saline at a dose of 2.5 ng/g of body weight. 

In both models, the mice were studied after different time periods (30, 45, and 60 min).

At the time of sacrifice, animals were anesthetized with 5 mg/kg xylazine (Rompun, Bayer AG, Leverkusen, Germany) and 35 mg/kg ketamine (Ketolar, Pfizer, New York City, NY, United States), and the kidneys were perfused in situ with cold saline before removal (n = 10 mice per group). A piece of the kidney (2/3) was fixed, embedded in paraffin, and used for immunohistochemistry, while another piece was incubated for 20 min in KHS-HEPES buffer pH 7.4 (130 mM NaC1, 5 mM KC1, 1.3 mMMgC12, 1.2 mM sodium phosphate, 0.5 mM EGTA, 10 mM glucose, 10 mM HEPES) containing 30% sucrose before being OCT-embedded.

### 2.2. CCN2 Fluorescence Cy5 Labeling and siRNA FAM Labeling

The C-terminal fragment of CCN2 (CCN2(IV); Preprotech) was resuspended in sterile water, and 0.5 μL of Tris-buffer (1 mol/L pH:8.8) was added to modify the pH of the protein. The fluorophore used to label the recombinant protein was Cy-5 (1 nmol/μL; Amersham, Loughborough, UK); the fluorophore was reconstituted in dimethylformamide (DMF) (80 pmol/mL; Sigma-Aldrich, Burlington, MA, USA) and incubated with CCN2(IV) recombinant protein for 30 to 45 min in darkness (protein fluor labeling reaction), and then, 1 μL of lysine 10 mmol/L was added to stop the reaction. Human EGFR siRNA (Ambion; Fisher Scientific; Waltham, MA, USA) used with this sequence was:Antisense (5′-3′): AUAUUCGUAGCAUUUAUGGag;Sense (5′-3′): CCAUAAAUGCUACGAAUAUtt.

The EGFR siRNA and the control siRNA were labeled with FAM fluorophore using Ambion’s Silencer siRNA Labeling kit Ambion; Fisher Scientific; Waltham, MA, USA) following the manufacturer’s instructions.

### 2.3. Confocal Microscopy Analysis of CCN2(IV) Binding In Vivo

Renal OCT-tissue sections (3 μm) were used for confocal microscopy studies. Images of kidney sections were captured using a Leica TCS SP5 confocal microscope (Leica Microsystems; Wetzlar, Germany) equipped with an X63 oil immersion lens. Fluorophore Cy-5-emitted fluorescence was monitored with a 550/20 nm band-pass filter or with a 670 nm long-pass filter, and 4′,6-diamidino-2-phenylindole (DAPI) was excited using a diode laser [34].

### 2.4. Kidney Activation of EGFR Signaling Pathway

Kidney sections were stained with anti-phosphorylated EGFR (1:200; Dako Agilent, Santa Clara, CA, USA), Alexa Fluor 488 conjugated goat anti-mouse secondary antibodies (1:300; Invitrogen, Waltham, MA, USA), and examined using a Leica DM-IRB confocal microscope.

### 2.5. Cell Cultures

Human renal proximal tubular epithelial cells (HK2 cell line, ATCC CRL-2190; Gaithersburg, MD, USA) were grown in RPMI 1640 medium (Lonza; Basel, Switzerland) with 10% heat-inactivated fetal bovine serum (FBS; Gibco, Billings, MT, USA); 2 mmol/L glutamine; 100 U/mL penicillin; 100 mg/mL streptomycin; 5 mg/mL insulin, transferrin, selenite (ITS) (Gibco, Billings, MT, USA); and 36 ng/mL hydrocortisone (Sigma-Aldrich; MA, USA) in 5% CO_2_ at 37 °C. Cells were cultured in six-well plates, and at 60–70% of confluence, cells were growth-arrested in serum-free medium for 24 h and treated with vehicle or recombinant CCN2(IV)-Cy5 (100 ng/mL, Peprotech, London, UK) for short periods of time in a serum-free medium. 

### 2.6. In Vitro Cy5-CCN2(IV) Binding to EGFR

HK2 cells were silenced using either a predesigned siRNA targeting the human EGFR or a non-specific control siRNA, both FAM-labeled (Ambion, Fisher Scientific, Waltham, MA, USA). Subconfluent HK2 cells were transfected for 24 h with LipofectamineTM RNAiMAX reagent (Invitrogen, Waltham, MA, USA) according to the manufacturer’s guidelines. Cells were then incubated in a serum-free medium for 24 h before the experiments.

### 2.7. Live-Cell Confocal Microscopy

Cells were treated with Cy5-CCN2(IV) and examined by a Leica DM-IRB confocal microscope (Leica Microsystems; Wetzlar, Germany) equipped with a X40 oil immersion objective. Nuclei were stained with 1 μg/mL DAPI (4′,6-diamidino-2-phenylindole; Sigma-Aldrich) to control for equal cell density. The absence of the primary antibody was used as a negative control. Fluorophore Cy-5 emitted fluorescence was monitored with a 550 ± 20 nm band pass or a 670 nm long-pass filter, and DAPI was excited using a DIODE laser. 

## 3. Results

### 3.1. CCN2 Binds to Tubular Epithelial Cells, but Not to Glomerular Cells, in a Model of Cy5-CCN2(IV) Intra-Renal Injection

To explore the CCN2 binding sites in vivo in the kidney, we developed an experimental murine model by direct renal injection of the recombinant CCN2(IV) protein, previously labeled with a fluorochrome. For these experiments, labeled Cy5-CCN2(IV) was injected in the right kidney, using its corresponding contralateral kidney as a control. The mice were sacrificed at different times, and then renal tissue was embedded in OCT, and the localization of CCN2(IV) binding to renal structures was done by confocal microscopy. In Cy5-CCN2 (IV)-injected kidneys, a red fluorescent signal was observed, mainly in tubular epithelial cells (Figure 1). The maximal fluorescence intensity signal was reached at 30 min, and no red signal was found in the contralateral kidneys (not shown). 

Interestingly, red immunofluorescence was only observed in the tubuli but not in other renal structures, such as the glomeruli (Figure 2).

### 3.2. CCN2 Binds to Tubular Epithelial Cells, but Not to Glomerular Cells, in a Model of Systemic Administration of Cy5-CCN2(IV) into Mice

To evaluate whether there are some differences between local and systemic administration of the recombinant CCN2(IV), a mice model of Cy5-CCN2(IV) i.p. injection was conducted. In the kidneys of Cy5-CCN2(IV)-i.p injected mice, a red immunofluorescence signal was found, showing the maximal intensity at 30 min (Figure 3). 

The Cy5-CCN2(IV) binding was mainly located in proximal tubular epithelial cells (Figure 4), as demonstrated using specific markers of proximal (LotusTetragonolobus Lectin, green) or distal (Dolichos Biflorus Lectin, green) tubuli. 

### 3.3. CCN2 Binding Is Linked to EGFR Pathway Activation in the Kidney

Our second aim was to evaluate whether intra-renal injection of CCN2(IV) could activate the EGFR signaling pathway. We have previously demonstrated that CCN2 is an EGFR ligand [34]. Ligand binding to EGFR induces a conformational change leading to the formation of receptor homo- or heterodimers and subsequent activation (phosphorylation of specific tyrosine (Tyr) residues) of the tyrosine kinase domain located in the cytoplasmic tail of the receptor [48]. In the model of renal Cy5-CCN2(IV) injection, we found that cells with positive Cy5-CCN2(IV) binding also presented phosphorylated-EGFR immunostaining (Figure 5). Moreover, co-localization of CCN2(IV) binding and EGFR activation in the same tubular cells were also observed in the model of systemic CCN2(IV) administration (not shown). All these data demonstrate that CCN2(IV) binds to EGFR and activates its downstream signaling pathway in tubular epithelial cells in vivo.

### 3.4. CCN2 Binding Is Linked to EGFR Pathway Activation in the Kidney

To further demonstrate CCN2(IV) binding to EGFR, cultured human tubular epithelial live-cell imaging was performed by confocal time-lapse microscopy. The involvement of EGFR was evaluated by gene silencing. HK2 cells were transfected with a siRNA against EGFR or its corresponding scrambled control siRNA (both FAM-labeled). After adding Cy5-CCN2(IV) to siRNA-transfected control HK2 cells, a red immunofluorescent signal was rapidly located in the cellular membrane (Figure 6), whereas in EGFR-silenced cells, the signal was decreased (Figure 6). As a control of the technique, Cy5-fluorochrome was added to the medium, showing no cellular binding (not shown). These data show that CCN2(IV) in vivo binds to tubular epithelial cells and activates the EGFR pathway in these cells.

## 4. Discussion

Our data demonstrate that CCN2(IV) preferentially binds to the tubular epithelium in the mouse kidney. Moreover, colocalization of EGFR pathway activation was found in the same tubular cells. These findings show that tubular epithelial cells are a key target of CCN2 effects in the kidney. 

Many in vitro studies in cultured renal cells, including tubular epithelial cells, have demonstrated that CCN2 and CCN2(IV) exert many biological functions, regulating cell growth, proliferation, and proinflammatory factors [7], which could contribute to fibrosis as shown by increased ECM production in mesangial cells [49,50] and kidney fibroblasts [51]. Our preclinical data of exogenous administration of the recombinant protein in mice show that CCN2(IV) preferentially binds to tubular epithelial cells and, therefore, may induce deleterious effects acting on these cells. Tubular epithelial cells are the main actors in the physiological kidney function, participating in protein reabsorption. To this aim, tubular cells require an elevated energy demand, presenting a high number of mitochondria. However, these cells are very vulnerable to damage, including hypoxia, toxics, or metabolic stress [52]. Many studies have shown that phenotype changes in tubular epithelial cells, including partial EMT or development of a senescence-like molecular signature associated with inflammation and fibrosis, can contribute to kidney damage progression [52]. Previous in vitro studies have shown that in cultured tubular epithelial cells, incubation with recombinant CCN2(IV) protein activates the NF-κB pathway and regulates several proinflammatory factors, including cytokines and chemokines [14]. Moreover, both CCN2 and CCN2(IV) activate tubular cells, leading to phenotype changes associated with partial EMT [12,53,54]. These EMT-induced changes are mediated by the activation of MAPK-ERK cascade [55,56], ERGF, JAK-STAT3, and NF-κB signaling pathways [21]. In addition, stimulation with CCN2(IV) also induces cell growth arrest of cultured tubular cells [21] and activation of senescent-related mechanisms [11]. Our findings show that CCN2(IV) preferentially binds to tubular epithelial cells in the mouse kidney. These data, together with the in vitro evidence of deleterious effects of recombinant CCN2 or CCN2(IV) in tubular cells, confirm the importance of this protein acting as a growth factor in the progression of kidney damage. 

In human and experimental kidney damage, upregulation of CCN2 gene expression and increased CCN2 protein production in glomerular and tubulointerstital areas have been described, showing the involvement of this factor in the progression of kidney diseases [7,19,57]. The complete CCN2 molecule can be cleaved by different proteases, releasing CCN2 from the ECM as well as producing several CCN2 biologically active fragments [58]. In this sense, CCN2(IV) can be released by MMP2 [59]. Several studies have measured circulating or urinary levels of CCN2 to evaluate whether this molecule could be a biomarker of disease progression, using antibodies that recognize the N-terminal or the C-terminal fragment [60,61]. This suggests that in pathological conditions, CCN2 overexpression not only contributes to elevated CCN2 content of this matricellular protein in the ECM but also CCN2 can be released to neighboring cells and to circulation, acting as a growth factor. However, an unresolved question is the local concentrations of CCN2 degradation products in pathological conditions. Our studies showing binding of the CCN2(IV) in tubular epithelial cells remark the importance of future studies evaluating the presence of CCN2 degradation products in kidney diseases.

In this paper, we demonstrated that CCN2-binding sites are mainly located in tubular epithelial cells, and upon CCN2 binding to these cells, EGFR signaling pathway is activated. Many studies using different strategies for blocking CCN2 activity have proven beneficial effects in experimental pathologies, including renal diseases [62,63]. Moreover, a phase 1 trial of the effects of an anti-CCN2 antibody in diabetic patients with microalbuminuria (indicative of early kidney changes) showed not only that the antibody was safe to use but it also reduced micro-albuminuria [64]. In addition, anti-CCN2 therapy is in phase 2 or 3 clinical trials in other diseases, including Duchenne muscular dystrophy (NCT02606136), idiopathic pulmonary fibrosis (NCT03955146), and pancreatic adenocarcinoma (NCT04229004). These data indicate that additional studies are needed to evaluate if CCN2 could be an anti-fibrotic target for CKD. On the other hand, EGFR blockers—including small kinase inhibitors such as erlotinib, gefitinib, or afatinib, or monoclonal antibodies—have been used in several human proliferative disorders, including lung cancer [65,66,67,68,69,70], opening novel opportunities to use EGFR blockers in renal diseases. Our findings confirm that CCN2 is linked to EGFR signaling activation in the kidney and support the importance of this receptor as a new therapeutic target for CKD. Nowadays, the current clinical treatment of CKD includes blockers of the renin angiotensin system and, more recently, SGLT2 inhibitors [71,72], which attenuate the proximal reabsorption of sodium. In fact, in vitro studies have found that SGLT2 inhibition prevents EMT-phenotypes changes in tubular cells by the suppression of sirtuin 3 and aberrant glycolysis [73], suggesting that targeting damage in this cell type could be a good therapeutic option for CKD. In summary, many data have found that in experimental and human CKD, there is an elevated CCN2 production in the kidney. Our data show that CCN2(IV) preferentially binds to tubular cells in the kidney; although they have clear limitations of preclinical studies using a pharmacological-type approach, they show that tubular cells are the main target cells of soluble CCN2, and therefore, CCN2 acting as a growth factor can bind to these cells to induce deleterious actions and contribute to renal damage progression.

## Figures and Tables

**Figure 1 biomolecules-12-00252-f001:**
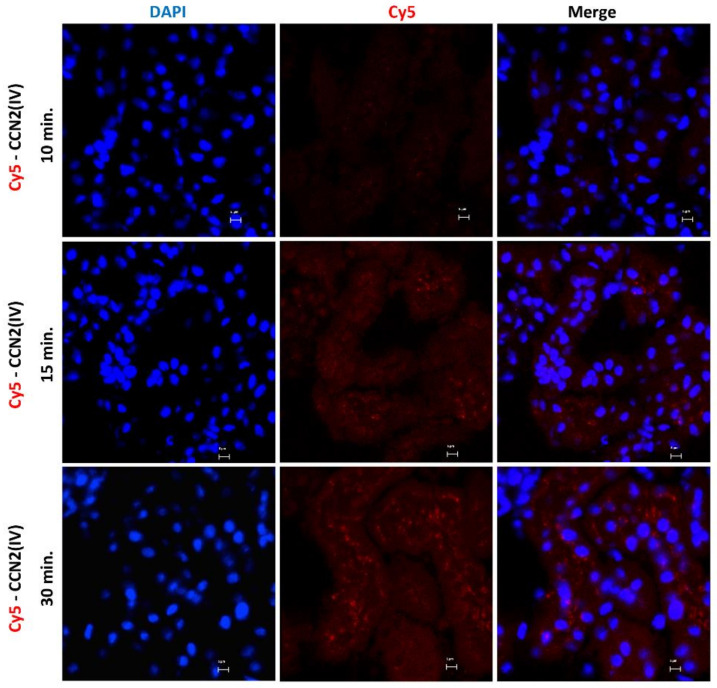
Evaluation of CCN2(IV) binding to renal tissue in the model of intra-renal injection. C57BL/6 mice were injected with Cy5-CCN2(IV) (dose of 2.5 ng/g of body weight, n = 3 mice per group) in the right kidney and sacrificed at different times (from 10 to 30 min). Frozen OCT-embedded renal samples were used for confocal microscopy. Red immunostaining (corresponding to CCN2(IV) labeled with Cy5 fluorophore) was found in tubular epithelial cells, showing CCN2(IV) binding to renal cells. Nuclei were stained with DAPI to distinguish renal structures. The overlaid images in red and blue (merge) indicated CCN2(IV) localization mainly in the membrane of tubuloepithelial cells. Figures show one representative mouse per group, and scale bar represents 5 μM.

**Figure 2 biomolecules-12-00252-f002:**
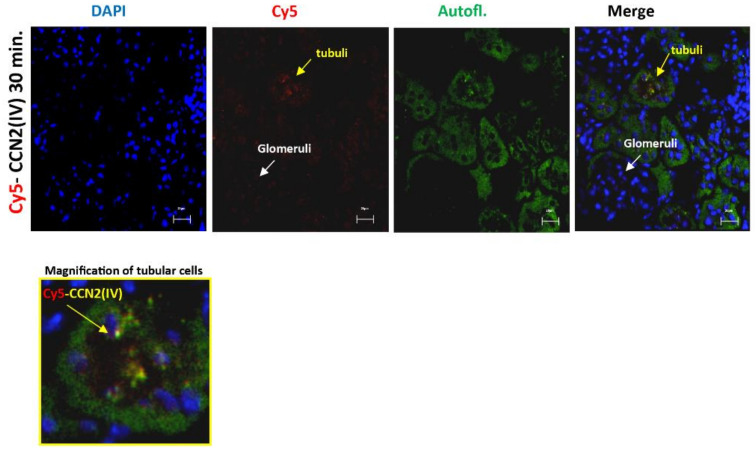
CCN2 (IV) binding was located in the tubuli but not in other renal structures in the kidney. Mice were injected with Cy5-CCN2(IV) in the right kidney and studied 30 min later. Red immunostaining was only found in the tubuli but was not present in the glomeruli. Renal tissue autofluorescence (green) and DAPI staining (blue) show renal structures. The overlaid images indicate CCN2(IV) localization only in the tubuli. Figures show a representative mouse per group, and scale bar represents 20 μM.

**Figure 3 biomolecules-12-00252-f003:**
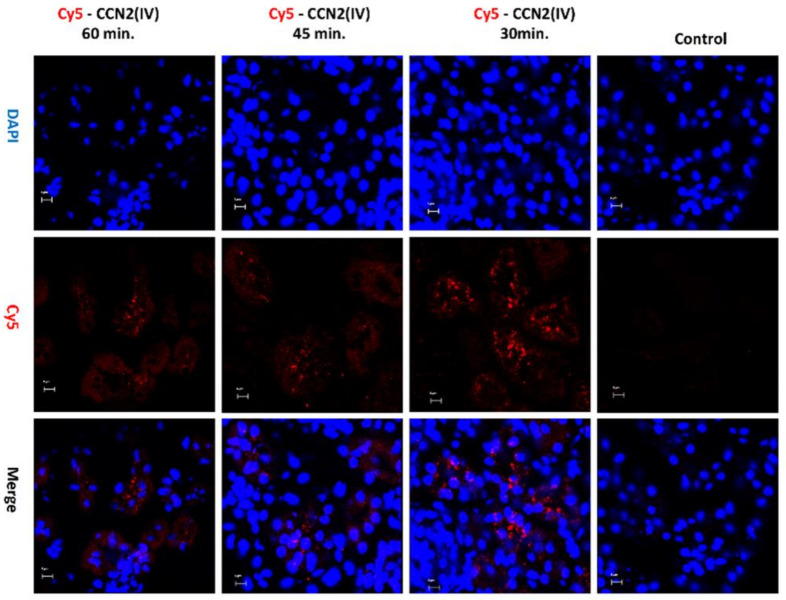
Evaluation of CCN2(IV) binding to the kidney in the model of systemic administration in mice. C57BL/6 mice were i.p. injected with 2.5 ng/g of body weight of recombinant Cy5- CCN2(IV) and sacrificed 30, 45, and 60 min later. In the kidneys from Cy5-CCN2(IV)-injected mice, red immunostaining corresponding to CCN2(IV) labeled with Cy5 fluorophore (red staining) was observed by confocal microscopy. Nuclei were stained with DAPI to locate renal structures. The overlaid images in red and blue (merge) indicated CCN2 (IV) localization mainly in the membrane of tubular epithelial cells. Figures show a representative mouse per group of 3 done, and scale bar represents 5 μM.

**Figure 4 biomolecules-12-00252-f004:**
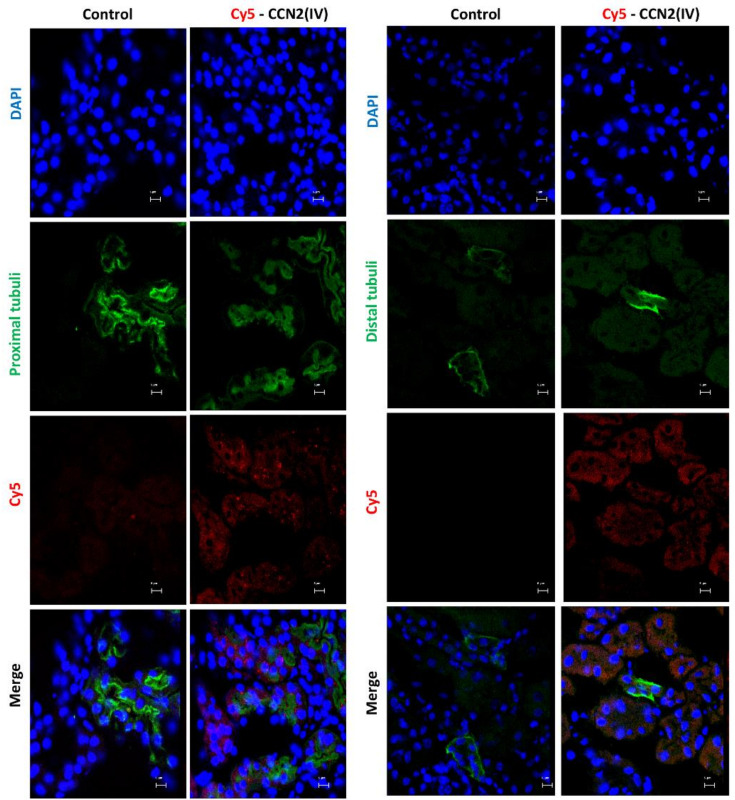
Determination of CCN2(IV) binding to proximal or distal tubuli in the kidney in the model of systemic administration of CCN2(IV). Co-localization of Cy5-CCN2(VI) (red) staining with markers of proximal (LotusTetragonolobus Lectin, green) or distal (Dolichos Biflorus Lectin, green) tubuli in CCN2(IV)-injected mice. Proximal tubuli are the main sites of Cy5-CCN2 binding sites. The scale bar represents 5 μM.

**Figure 5 biomolecules-12-00252-f005:**
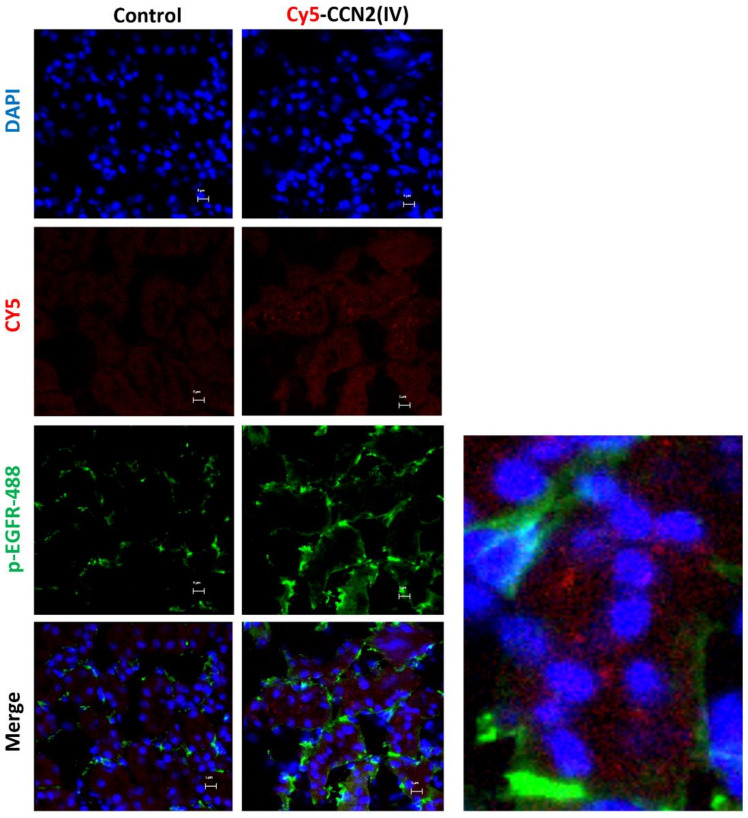
Localization of CCN2(IV) binding and EGFR activation in tubuloepithelial cells in vivo. C57BL/6 mice were injected with CCN2(IV)-Cy5 (2.5 ng/g of body weight, n = 3 mice per group) in the right kidney, using its corresponding contralateral kidney as control and sacrificed at 30 min. Figure shows Cy5-CCN2(IV) binding in red, activated EGFR in green (corresponding to phosphorylated EGFR (Alexa fluor^®^ 488)), and nuclei in blue (DAPI). The overlaid images show the colocalization of CCN2(IV) with EGFR activated in several tubule epithelial cells. A magnification is shown on the right. Figures show one representative mouse per group, and scale bar represents 5 μM.

**Figure 6 biomolecules-12-00252-f006:**
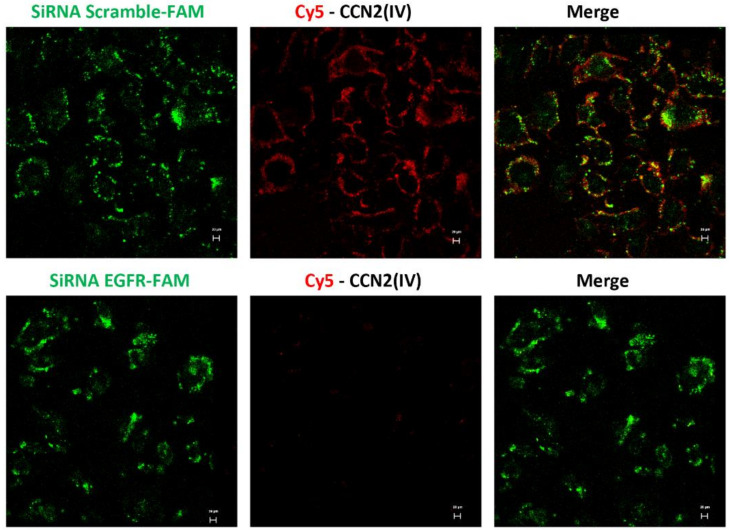
In vitro binding of Cy5-CCN2(IV) to tubular epithelial cells via EGFR. Serum-starved human tubular epithelial cells were transfected with a siRNA against EGFR or its corresponding scramble control siRNA, both FAM-labeled (green) and then stimulated with Cy5-CCN2(IV) for 10 min. Only cells expressing EGFR can bind to Cy5-CCN2(IV) (red signal), mainly located in the cellular membrane, whereas this signal is not found in EGFR-silenced cells. The scale bar represents 20 μM.

## Data Availability

Not applicable.

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
