# Peer review of "CCN2 Binds to Tubular Epithelial Cells in the Kidney"

_biomolecules, 2022, doi:10.3390/biom12020252_

Round 1

Reviewer 1 Report

In Material and Methods please describe in detail the rates of the sacrifice of the animals, due to dynamics of CCL2 and EGFRr expression in the kidney. 

Author Response

Answer to Reviewer 1

In Material and Methods please describe in detail the rates of the sacrifice of the animals, due to dynamics of CCN2 and EGFRr expression in the kidney. 

Thank you for your suggestion, these data have been implemented in the  Material and Methods section.

Reviewer 2 Report

Summary

The study from Rayego-Mateos et al. investigates the role of Cellular communication network-2 (CCN2), or connective growth factor (CTGF) in the kidney. The matricellular protein CCN2 plays a role as a growth factor and as a fibrotic marker. CCN2 expression is upregulated by TGF-b, angiotensin II, increased glucose levels, cellular stress and in kidney damage and CCN2 overexpression associated with cellular proliferation and extracellular matrix accumulation in both glomerular and interstitial areas. In the kidney 2 potential receptors have been identified; TrkA and EGFR.

In their previous study, the group identified the C-terminal module of CCN2 to bind to EGFR using surface plasmon resonance analysis.

Specifically, this study investigates the interaction between CCN2 and EGFR in vivo. CCN2 contains 4 functional modules and here they present the interaction of the carboxyl-terminal cysteine knot module 4 (CCN2 (IV)) with EGFR. For this, CCN2 (IV) was fluorescently labelled and injected into the right kidney or injected into the peritoneum and localisation in the kidney was examined by immunofluorescence at different time points post-injection. To investigate EGFR activation by CCN2(IV), the authors examined phosphorylated EGFR in vivo. Finally, to investigate CCN2(IV) binding to EGFR, the authors knockdown EGFR in vitro and analysed CCN2(IV) localisation to the cells.

Overall:

Overall, the aim of the study by Rayego-Mateos et al. is of great interest for the research community. As pointed out by the authors, CCN2 is a potential fibrotic biomarker in kidney disease and therefore a potential therapeutic target. Characterisation of its target cells and signaling is an important part in the development of new therapeutics. The authors demonstrate CCN2 binding to tubular epithelial cells upon injection of fluorescently labelled CCN2(IV) into the right kidney or systemically. For this, the authors analyse protein localisation by immunofluorescence in the mouse kidney. Whilst the authors demonstrate localisation of Cy5-CCN2(IV) to the tubules, it is unclear whether this binding is specific. To exclude the possibility of unspecific binding/ uptake of Cy5, the experiments should be performed using Cy5 only.

Instead, the authors inject Cy5-CCN2(IV) into the right kidney and also into the peritoneum. The relevance of the two different routes of administration is unclear.

One of the authors main aims was to study whether intra-renal injection of CCN2(IV) could activate EGFR signalling and whether CCN2(IV) binds to EGFR. Provided evidence for CCN2(IV) binding to EGFR comes from their previous study performing SPR. Co-labelling experiments in this study support localisation of both proteins to the same cells but not in close proximity to support the idea of direct binding.

Specific points:

Figure 1: The authors inject CY5-labelled CCN2(IV) into the right kidney of mice and find localisation of Cy5-CCN2(IV) in tubular epithelial cells (in the membranes) with the highest signal detectable after 30 minutes (analysed 10, 15, 30 minutes). No quantification of the immunofluorescence was performed and the scale bar is missing.

Figure 2: The authors describe specific localisation of Cy5-CCN2 (IV) to the tubules and absence from glomeruli at 30 minutes post-injection.

Transient localisation of Cy5-CCN2(IV) in glomeruli should be characterised at 10 minutes and 15 minutes post-injection as it is expected for the labelled CCN2(IV) construct to pass through the glomeruli to reach the tubules. Furthermore, co-labelling with a glomerular marker should be performed. Scale bar is missing.

Figure 3: The aim is to evaluate whether there are differences between local and systemic administration. Therefore, the authors inject Cy5-CCN2(IV) intraperitoneally. Cy5 immunofluorescence is analysed after 30, 45, 60 minutes and the authors describe highest immunofluorescent signal in the tubular cells after 30 minutes. No tubular marker has been included and scale bar is missing. Glomerular localisation was not studied.

Figure 4: The authors investigate the tubular localisation of Cy5-CCN2(IV) in more detail and perform co-labelling for proximal and distal tubule markers. The authors describe Cy5-CCN2(IV) binding mainly to the proximal tubules. However, no quantification is provided and the Cy5 staining in the distal tubules does not look convincing. Whilst Cy5-CCN2(IV) appeared punctuated along the membrane in previous images, the signal shown here appears uniform and of low intensity. The merged images are overpowered by the DAPI stain and therefore the DAPI channel should be removed from the merged images. Scale bar is missing.

Figure 5: The aim of this figure was to investigate whether Cy5-CCN2(IV) can activate EGFR signalling. Here, the authors present localisation of Cy5-CCN2(IV) and phosphorylated-EGFR in the same tubules. The quality of the magnified images is too low and images should be re-taken using a different objective. Whilst the authors describe CCN2(IV) as a direct ligand of EGFR, the immunofluorescence presented in this figure does not support this theory. Whilst Cy5-CCN2(IV) localises to the luminous, apical side of the tubules, phosphorylated EGFR staining is observed at the basal side of the tubules.

Figure 6: Next, the authors further investigate CCN2(IV) binding to EGFR in vitro. For this, HK2 cells were transfected with siRNA targeting EGFR transcript, followed by addition of CY5-labelled CCN2(IV) to the medium for 10 minutes and examination of CY5-labelled CCN2(IV) to the cells. Evidence of knockdown by the used siRNA is lacking. Staining for the transfected siRNA is not sufficient to draw any conclusions on involvement of EGFR in the localisation of CCN2(IV). No quantification is presented, nor a scale bar.

Minor points:

Page 2, line 89: please replace “have showed” with “have shown”

Page 8, line 274: “control data not shown”. It would be helpful if control experiments were included in the supplement.

Page 9, line 286: please remove “colocalisation of”, as this could be misleading.

Page 9, line 290: please replace “demonstrate” with “demonstrated”

Page 10, line329: “All these data confirm that EGFR is a functional CCN2 receptor in the murine kidney”, I suggest to remove the sentence as not sufficient evidence has been presented to support this statement.

Page 10, line 352: please remove “(via EGFR)”

Material and Methods: Could the authors please include the siRNA sequence for the EGFR knockdown experiments.

Author Response

The answer to the reviewer 2 is attached as PDF below.

Reviewer 3 Report

Subject Appropriateness of the Manuscript:

The topic of this manuscript falls within the scope of Biomolecules

Recommendation

Accept (minor revision)

Comments

The paper entitled “CCN2 binds to tubular epithelial cells in the kidney” (biomolecules-1558397) is an interesting, clearly written work concerning the potential role and molecular mechanisms by which cellular communication network-2 (CCN2), also called connective tissue growth factor (CTGF), can be considered as fibrotic biomarker and a growth factor involved in kidney damage progression. The aim of their study was to localize CCN2 binding in vivo in the kidney, using the C-terminal module CCN2(IV) labeled with a fluorophore (Cy5) and confocal microscopy analysis. For this purpose, both intraperitoneal and direct intra-renal injection of Cy5-CCN2(IV) in mice were employed. The study demonstrated that CCN2(IV)-binding sites are mainly located in tubular epithelial cells, while no signal was detected in glomeruli. Moreover, co-localization of Cy5-CCN2(IV) binding and activated EGFR was found in tubules, confirming that CCN-2 is linked to EGFR signaling activation in the kidney. The Authors conclude that EGFR is a functional receptor for CCN2 and can be considered as a new therapeutic target for chronic kidney diseases. The study is well designed, and the subject is highly relevant.

Some of my minor concerns are:

  1. The Authors should discuss why CCN2(IV) preferentially binds to the tubular epithelial cells. This is important difference from other studies, especially since, on the basis of the data available so far, it is believed that CCN2 plays an important role in the development of glomerular and tubulointerstitial fibrosis in progressive kidney diseases.
  2. The information about the limitations of this study should be added into the last paragraph of the Discussion section.

Author Response

Comments

The paper entitled “CCN2 binds to tubular epithelial cells in the kidney” (biomolecules-1558397) is an interesting, clearly written work concerning the potential role and molecular mechanisms by which cellular communication network-2 (CCN2), also called connective tissue growth factor (CTGF), can be considered as fibrotic biomarker and a growth factor involved in kidney damage progression. The aim of their study was to localize CCN2 binding in vivo in the kidney, using the C-terminal module CCN2(IV) labeled with a fluorophore (Cy5) and confocal microscopy analysis. For this purpose, both intraperitoneal and direct intra-renal injection of Cy5-CCN2(IV) in mice were employed. The study demonstrated that CCN2(IV)-binding sites are mainly located in tubular epithelial cells, while no signal was detected in glomeruli. Moreover, co-localization of Cy5-CCN2(IV) binding and activated EGFR was found in tubules, confirming that CCN-2 is linked to EGFR signaling activation in the kidney. The Authors conclude that EGFR is a functional receptor for CCN2 and can be considered as a new therapeutic target for chronic kidney diseases. The study is well designed, and the subject is highly relevant.

Some of my minor concerns are:

  1. The Authors should discuss why CCN2(IV) preferentially binds to the tubular epithelial cells. This is an important difference from other studies, especially since, on the basis of the data available so far, it is believed that CCN2 plays an important role in the development of glomerular and tubulointerstitial fibrosis in progressive kidney diseases.

As the reviewer indicates, CCN2 participates in CKD, and different studies (including ours) have demonstrated that CCN2 is overexpressed in different kidney cell types (depending on the pathology). However, in this study, we focus on the effect of CCN2(IV) on the kidney physiological conditions. Accordingly, some parts of the discussion have been modified.

  1. The information about the limitations of this study should be added into the last paragraph of the Discussion section.

Thank you for your suggestion, we include a paragraph about the limitations of the study at the end of the discussion section

Round 2

Reviewer 2 Report

Thank you for the answers provided. Most of my concerns have been resolved. Whilst scale bars have been included, it is very hard to read them still. Would the authors mind indicating the scale bar in the figure legends as well please.

Author Response

As referee recommends we include the reference to scale bars in the footnotes.